# Anti-Pathogenic Properties of the Combination of a T3SS Inhibitory Halogenated Pyrrolidone with C-30 Furanone

**DOI:** 10.3390/molecules26247635

**Published:** 2021-12-16

**Authors:** Nelly Araceli Aburto-Rodríguez, Naybi Muñoz-Cázares, Víctor Alberto Castro-Torres, Bertha González-Pedrajo, Miguel Díaz-Guerrero, Rodolfo García-Contreras, Héctor Quezada, Israel Castillo-Juárez, Mariano Martínez-Vázquez

**Affiliations:** 1Departamento de Productos Naturales, Instituto de Química, Universidad Nacional Autónoma de México, Circuito Exterior, Ciudad Universitaria, Coyoacán, Ciudad de Mexico 04510, Mexico; nellyaraceli@ciencias.unam.mx (N.A.A.-R.); lrvictor_wilde@hotmail.com (V.A.C.-T.); 2Laboratorio de Fitoquímica, Posgrado de Botánica, Colegio de Postgraduados, Texcoco 56230, Mexico; munoz.naybi@colpos.mx; 3Departamento de Genética Molecular, Instituto de Fisiología Celular, Universidad Nacional Autónoma de México, Ciudad de Mexico 04510, Mexico; bpedrajo@ifc.unam.mx (B.G.-P.); madiaz@ifc.unam.mx (M.D.-G.); 4Departamento de Microbiología y Parasitología, Facultad de Medicina, Universidad Nacional Autónoma de México, Ciudad Universitaria, Coyoacán, Ciudad de Mexico 04510, Mexico; rgarc@bq.unam.mx; 5Laboratorio de Investigación en Inmunología y Proteómica, Hospital Infantil de México Federico Gómez, Ciudad de Mexico 06720, Mexico; hquezadap@yahoo.com.mx

**Keywords:** anti-virulence, pyrrolidone, type III secretion system, quorum sensing, anti-virulence combination therapies

## Abstract

Antimicrobial resistance is one of the current public health challenges to be solved. The World Health Organization (WHO) has urgently called for the development of strategies to expand the increasingly limited antimicrobial arsenal. The development of anti-virulence therapies is a viable option to counteract bacterial infections with the possibility of reducing the generation of resistance. Here we report on the chemical structures of pyrrolidones DEXT 1–4 (previously identified as furan derivatives) and their anti-virulence activity on *Pseudomonas aeruginosa* strains. DEXT 1–4 were shown to inhibit biofilm formation, swarming motility, and secretion of ExoU and ExoT effector proteins. Also, the anti-pathogenic property of DEXT-3 alone or in combination with furanone C-30 (quorum sensing inhibitor) or MBX-1641 (type III secretion system inhibitor) was analyzed in a model of necrosis induced by *P. aeruginosa* PA14. All treatments reduced necrosis; however, only the combination of C-30 50 µM with DEXT-3 100 µM showed significant inhibition of bacterial growth in the inoculation area and systemic dispersion. In conclusion, pyrrolidones DEXT 1–4 are chemical structures capable of reducing the pathogenicity of *P. aeruginosa* and with the potential for the development of anti-virulence combination therapies.

## 1. Introduction

Multiple reports in the specialized literature indicate that bacterial multidrug resistance (MDR) is an emerging global public health problem. The World Health Organization (WHO) announced a list of priority microbes that need to be fought due to their high resistance rates, and *Pseudomonas aeruginosa* is one of the main ones considered critical [1]. This bacterium is the most common Gram-negative pathogen that causes nosocomial pneumonia. For example, in a pediatric hospital in Mexico, 46 nosocomial infections caused by *P. aeruginosa* were detected from 2007 to 2013, with a mortality rate of 17.39% [2].

The pathogenicity of *P. aeruginosa* can be explained by the production of several virulence factors such as exoproteases, toxins, pyocyanin, rhamnolipids, and swarming motility regulated by quorum sensing systems (QSS) [3]. Inhibition of QSS in *P. aeruginosa* has been reported to reduce virulence factor production and pathogenicity in various in vivo models, including murine models [3,4]. Currently, QSS inhibition is presented as an alternative to eliminate infections caused by MDR bacteria [5]. Furthermore, because this therapy does not exert direct selection pressure, it may have a low probability of developing resistance [6,7].

In addition, *P. aeruginosa* infections can be eradicated with an adequate treatment of antibiotics in its planktonic state; however, it is challenging to do so when they form biofilm [8].

The earliest QSS inhibitors were synthesized using the 2-furanone structural template, the essential residue of acylated homoserine lactones (AHL). Among the anti-QSS furanones, C-30 (5*Z*)-4-Bromo-5-(bromomethylene)-2(5*H*)-furanone) is one of the most relevant [9,10]. C-30 has activity against several virulence factors, which is why it has become the usual reference compound in evaluating new anti-QSS molecules [9]. Besides, in in vivo studies, C-30 decreases the bacterial population in lung infection in mice [10] and inhibits QS of some *P. aeruginosa* clinical isolates [11].

Another essential mechanism that allows the proliferation of *P. aeruginosa* in infected hosts and to defeat host defenses is the export of various virulence proteins through specialized secretory machinery. This bacterium uses a type III secretion system (T3SS) to inject effector proteins directly into the contacted host cell [12]. Once inside the host cell, effector molecules modulate essential host functions such as cytoskeletal organization and signal transduction [13]. Four different type III secreted effectors, ExoS, ExoT, ExoY, and ExoU have been identified [14].

Several compounds with different structural backbones have been evaluated as bactericidal agents. Among them are natural products with a pyrrolidone residue in their structure [15]. For example, from the fermentation broth of the bacterium *Rapidithrix thailandica*, a new antibacterial phenyl pyrrolidone derivative was isolated [16]. The so-called penicillenols isolated from *Aspergillus restrictus*, which include a pyrrolidone residue in their structures, inhibit biofilm formation and eradicate pre-developed biofilms of *Candida albicans* [17]. Additionally, pyrrolidone dithiocarbamate has been reported to function as an extremely potent compound against infection by human rhinoviruses and poliovirus in cell culture [18].

This work reports the inhibition of biofilm production, swarming motility, and type III secretion of *P. aeruginosa* by halogenated pyrrolidone derivatives. Also, the antipathogenic effect of a bromopyrrolidone DEXT-3 derivative is reported in a mouse abscess model.

## 2. Results

### 2.1. Synthesis of Halogen Pyrrolidones (DEXT)

The synthesis of pyrrolidones (DEXT 1–4) was achieved through a one-pot reaction between diethyl acetylene dicarboxylate, aromatic amines and aromatic aldehydes, as previously reported for the synthesis of similar structures (Figure 1) [19].

### 2.2. Anti-Virulence Properties of Halogenated Pyrrolidones

The pyrrolidones DEXT 1–4 did not show a bactericidal effect at 10, 50, and 100 μM concentrations on the INP-42 strain; however, DEXT-3 at 100 μM induced a slight growth delay in the *P. aeruginosa* PA14 and INP-57M strains (Appendix A). Similarly, the DEXT 1–4 compounds did not significantly reduce pyocyanin production. However, DEXT-1, DEXT-2, and DEXT-4 inhibited protease activity by 10 to 30% in the two clinical isolates INP-42 and INP-57M (Appendix A).

All DEXT 1–4 derivatives showed significant swarming inhibitory activity in the three strains evaluated (Figure 2). Likewise, nearly all pyrrolidones significantly reduced biofilm formation in a dose-response manner in the 30 to 80% range; only DEXT-4 was inactive on the INP-57M clinical isolate (Figure 3).

### 2.3. Inhibition of Type III Effector Secretion by Halogenated Pyrrolidones

The type III protein secretion assay was performed as described in Materials and Methods. Briefly, secreted proteins from bacteria treated with DEXT 1–4 pyrrolidones were recovered from the supernatant by TCA-precipitation at 4 °C followed by centrifugation. Type III secreted proteins were separated on a 15% SDS-PAGE, transferred to a nitrocellulose membrane, and probed against anti-ExoU and anti-ExoS polyclonal antibodies. The anti-ExoS antibodies cross-react with ExoT. Our results showed that the DEXT 1–4 pyrrolidones inhibited the secretion of ExoT and ExoU effectors (Figure 4A) without affecting bacterial growth (Figure 4B).

### 2.4. DEXT-3 Antipathogenic Activity in a Murine Infection Model

The model of generation of necrosis by *P. aeruginosa* in mice was used to evaluate the antipathogenic effect in vivo (Figure 5). For these experiments, DEXT-3 was selected because it has a better dissolution in DMSO (vehicle) and showed good anti-virulence activity in vitro (Figure 2, Figure 3 and Figure 4). Furanone C-30 (QS inhibitor) and MBX-1641, a T3SS inhibitor, were used as positive controls. Also, combinations between T3SS and QS inhibitors were evaluated to determine the presence of synergisms.

Intramuscular inoculation of *P. aeruginosa* induced a necrotic area of 24.4 mm^2^ at 48 h and 16 mm^2^ at 72 h (Figure 5A,B). Coadministration of C-30 50 µM reduced it to 11.1 mm^2^, and with MBX-1641 to 8.2 mm^2^. DEXT-**3** 100 µM showed the best effect, reducing the necrotic area to 6.3 mm^2^, which was significantly enhanced in combination with C-30 50 to 2.0 mm^2^ at 48 h (α = 0.05) and 1.4 mm^2^ at 72 h (α = 0.05) (Figure 5A,B).

At 72 h, the establishment of the bacteria in the inoculation area and the systemic dispersion were determined by quantifying *P. aeruginosa* in lesions and livers of the surviving animals. In the untreated control group, bacteria in the inoculation area were log_10_ 8.7 CFU/g, and in the liver, log_10_ 6.3 CFU/g (Figure 5C). However, in all treatments, the bacterial load in lesions was reduced (log_10_ 5.2–6.4 CFU/g and in liver of log_10_ 0.09–0.19 CFU/g), only with MBX-1641 200 µM and DEXT-**3** 100 µM was dispersion to the liver avoided (Figure 5C). At the same time, the combination of C-30 50 µM + DEXT-**3** 100 µM was the only one that significantly reduced the establishment to log_10_ 1.1 CFU/g (α = 0.05) (Figure 5C).

Finally, *P. aeruginosa* infection killed 10% of the animals at 24 h and 30% from 48 to 92 h (Figure 5D). In the case of DEXT-**3** 100 µM, a survival of 80% was recorded, which in combination with C-30 50 µM increased to 100% (Figure 5D).

## 3. Discussion

Continuing with our systematic research on the biological properties of 2-(5*H*)-furanone derivatives, we recently published the synthesis and cytotoxic evaluation of new halogen-4-alkyl-5-phenyl-3-(phenylamino)-furan-2-(5*H*)-one-type derivatives [19]. These furanones were synthesized through a one-pot reaction between diethyl acetylene dicarboxylate, aromatic amines, and aromatic aldehydes, as reported for the **5** and **6** syntheses (Figure 6). The structures of the synthesized compounds were elucidated by the conventional spectroscopic analysis (^1^H, ^13^C NMR, IR, UV, Mass Spectra) as well as by comparison of their spectroscopic and chemical data with those published in the literature [20].

To conduct studies on structure-activity relations, we decided to perform a structural determination through X-ray crystallographic analysis of furanone **6**. Unexpectedly, the result showed that the correct structure of this compound corresponded to the pyrrolidone DEXT-**2** (Figure 7).

A bibliographic review showed that DEXT-2 was previously synthesized from Cl-benzaldehyde, aniline, and diethyl acetylene carboxylate under microwave irradiation in the presence of *p*-TSOH [21]. Also, it was obtained from the same reactive as above, treated with tetragonal nano-ZrO_2_ particles as a reusable catalyst [22]. Furthermore, more recently, other halogen pyrrolidone derivatives were synthesized from diethyl acetylene dicarboxylate, aromatic amines, and aromatic aldehydes via metal-free photo-redox catalysis under rose bengal irradiation [23]. Moreover, additional pyrrolidone derivatives have been synthesized from the reaction of 3,4 difluoro aniline, diethyl acetylene dicarboxylate, and Cl-benzaldehyde in water and catalyzed with ZnO-[DABCO(C2COOH)2] 2 [24]. These results demonstrate that the previously published compounds as halogen-4-alkyl-5-phenyl-3-(phenylamino)-furan-2-(5*H*)-one-type derivatives [19] correspond to the pyrrolidone derivatives DEXT 1–4 and highlights the difficulty to discriminate between the pyrrolidone structures 1-4 and the previously reported furanones using conventional spectroscopy. The ^1^H, ^13^C NMR spectra of DEXT 1–4 derivates and their HQC and HMBC experiments are provided in the Supplementary Information. These results suggest reviewing the structures obtained by the method reported by Narayana [20].

Various synthetic procedures have been described for obtaining halogen pyrrolidones from aromatic amines and aldehydes in the presence of dialkyl acetylene dicarboxylates. However, most of these syntheses require acidic conditions, light irradiation, or ZnO complexes. On the contrary, in addition to the dimethyl acetylene carboxylate, amine, and aldehyde aromatic, our synthesis only used β-dextrin as a reusable catalyst and water as solvent. All the reactions listed in Figure 6 are clean and high yielding (85–94%).

The violacein pigment synthesized by *Chromobacterium violaceum* has shown bactericidal activity against *P. aeruginosa* [25]. Considering that the violacein structure contains a 2-pyrrolidone moiety, we decided to evaluate *P. aeruginosa* biofilm inhibition by DEXT-1-4 pyrrolidones.

Our findings showed that in *P. aeruginosa* PA14, INP-57M, and INP-42 strains, DEXT 1–4 derivatives showed statistically significant dose-dependent inhibition of biofilm production (Figure 3).

The T3SS mediates the delivery of virulence proteins called effectors directly into the host cells’ cytoplasm. Bacteria utilizing this system can directly inject bacterial proteins called effectors into host cells across bacterial and host membranes. These effectors affect multiple host cell functions, allowing the invading pathogen to colonize, multiply, and chronically persist in the host [26]. In *P. aeruginosa* the effectors can be divided into two groups. The cytotoxic strains such as PA103 or *P. aeruginosa* PA14 possess ExoU, ExoT, and sometimes ExoY, whereas invasive isolates such as PA01 or PAK possess ExoS, ExoT, and often ExoY [27]. The different collection of T3SS effectors denotes the possible phenotype of the strain during disease. For example, ExoU-producing strains cause rapid necrotic cell death, while ExoS-producing strains are internalized, resulting in delayed cell death [28]. Considering the above, we decided to evaluate the ExoU inhibition in *P. aeruginosa* PA14 strain by DEXT 1–4 pyrrolidones. Our results showed that DEXT 1–4 derivatives inhibited the secretion of ExoU and ExoT. Although DEXT-1, which did not have halogen substituents, was less active in the secretion inhibition, it also impeded biofilm production (Figure 3 and Figure 4).

As mentioned above, the presence of ExoU engages in necrotic processes. Therefore, it could be expected that the administration of any pyrrolidone DEXT 1–4 that inhibits the translocation of this effector to the target cell could inhibit, at least in part, the necrosis in an in vivo model of infection. The mouse abscess model was used to investigate this hypothesis using the pyrrolidone DEXT-3 as an inhibitor of ExoU secretion and *P. aeruginosa* PA14 strain as the inducer agent of necrosis. The selection of DEXT-3 was based on its better dissolution in DMSO compared to the other pyrrolidones. In addition to DEXT-3, both furanone C-30 (QS inhibitor) and MBX-1641 (T3SS synthetic inhibitor) were evaluated.

*P. aeruginosa* PA14 subcutaneous injection leads to the formation of an abscess that reached a maximum size of necrosis at two days after inoculation (Figure 5A). Mice inoculated with *P. aeruginosa* PA14 were treated with C-30, DEXT-3, or C-30 (50 μM) + MBX-1641 (200 μM), and the combination showed significantly smaller necrotic lesions. However, the C-30 (50 μM) + DEXT-3 (100 μM) combination was the most effective treatment (Figure 5B). On the other hand, inoculated mice treated with C-30, DEXT-3, or the C-30 (50 μM) + MBX-1641 (200 μM) combination exhibited a sizeable bacterial load on the lesion and liver site (5C). However, those treated with C-30 (50 μM) + DEXT-3 (100 μM) combination showed no dissemination in the liver and a lower bacterial load in the abscess (Figure 5C). It is essential to mention that for the effect of treatments to be representative, the inoculum should not be greater than 1 × 10^10^.

The administration of C-30 furanone or the C-30 (50 µM) + DEXT-3 (100 µM) combination increased survival in mice. In fact, with the later treatment, no death was registered. In contrast, the combination C-30 (50 µM) + MBX-1641 (200 µM) showed inhibition in necrosis formation in the abscess model and exhibited higher mortality than untreated mice. These findings indicate that although both combinations share the occurrence of C-30 at the same doses, the combination of T3SS inhibitor MBX-1641 with the anti-QSS C-30 induces mice toxicity (5D).

To eliminate the possibility that the protective effect shown by the treatments could be due to a bactericidal effect, DEXT-3 or MBX-1641 were evaluated as bactericidal agents. The results showed that they did not show a bactericidal effect.

## 4. Conclusions

Through the X-ray analysis of the single crystal, it was possible to determine the structure of DEXT-2 as a pyrrolidone derivative, allowing the correct labeling of the DEXT 1–4 compounds previously reported as furanone derivatives. The compounds were obtained with high yields through reactions in a single step and using water as a solvent, thus the synthetic process was improved compared to previous reports. DEXT 1–4 have anti-virulence properties related to biofilm inhibition and swarming motility in various strains, such as the reference strain and two clinical isolates, one of them multidrug-resistant. Furthermore, to the best of our knowledge, this is the first report of their T3SS inhibitory properties. Finally, the anti-pathogenic properties of DEXT-3 were corroborated in a murine infection model with *P. aeruginosa* PA14. Remarkably, the anti-pathogenic effect increases when combined with a QS inhibitor such as furanone C-30. This result indicates the potential of developing combination therapies using molecules with different anti-virulence targets.

## 5. Materials and Methods

### 5.1. Synthesis of Pyrrolidones DEXT 1–4

The general synthesis procedure for DEXT 1–4 was performed as follows (See Appendix A for spectroscopic data). β-cyclodextrin (0.226 g, 0.2 mmol) was dissolved in distilled water (30 mL). The solution was stirred for 10 min until a clear solution was obtained. To form aniline-cyclodextrin complex, an aniline derivative (2.0 mmol) was added and stirred for another 5 min. Subsequently, diethyl acetylenedicarboxylate (0.170 g, 2.0 mmol) was added slowly through an addition funnel, followed by an aromatic aldehyde (2.0 mmol), and the reaction mixture was heated at 60–70 °C. The reaction was monitored by thin-layer chromatography. The reaction blend was cooled at room temperature and extracted with ethyl acetate (4 × 10 mL). The organic layers were washed with water, saturated brine solution (2 × 10 mL) and dried over anhydrous Na_2_SO_4_. The combined organic layers were evaporated under reduced pressure. The resulting crude product reaction was purified by silica open column chromatography using ethyl acetate and hexane (7:3) mixture as eluent to give the respective furanones. The time and yield for each reaction are shown in Table 1. The pyrrolidones were dissolved in DMSO and stored at −20 °C.

*Ethyl 4-hydroxy-5-oxo-1,2-diphenyl-2,5-dihydro-1H-pyrrole-3-carboxylate* (DEXT-1). Yield (80%) as a pale-yellow solid, m.p. = 152–153 °C. HPLC purity: 99%. ^1^H NMR (300 MHz, CDCl_3_) δ 7.51–7.43 (m, 2H, CH-14-14′), 7.30–7.18 (m, 7H, CH-10-10′’, CH-11-11′, CH-13-13′), 7.12–7.02 (m, 1H, CH-12), 5.73 (s, 1H, CH-5), 4.18 (q, *J* = 7.1Hz, 2H, CH_2_-7), 1.17 (t, *J* = 7.1 Hz, 3H, CH_3_-8), ^13^C NMR (75 MHz, CDCl_3_) δ 165.22 (C2), 163.02 (C6), 156.58 (C13), 136.40 (C4), 135.22 (C9), 129.06 (C15-15′), 128.69 (C11-11′), 128.62 (C12), 127.64 (C10-10′), 125.94 (C16), 122.40 (C14-14′), 113.29 (C3), 61.69 (C5), 61.34 (C7), 14.02 (C8). MS, *m*/*z* (DART+): 324.12214, Anal. Calcd for ^12^C_19_^1^H_18_^14^N_1_^16^O_4_.

*Ethyl 2-(4-chlorophenyl)-4-hydroxy-5-oxo-1-phenyl-2,5-dihydro-1H-pyrrole-3-carboxylate* (DEXT-2). Yield (90%) as a pale-yellow solid, m.p. = 165–167 °C. HPLC purity: 99%. ^1^H NMR (300 MHz, CDCl_3_) δ 7.44 (dd, *J* = 8.6, 0.9 Hz, 2H, CH-14-14′), 7.30–7.26 (m, 2H, CH-15-15′), 7.24–7.21 (m, 2H, CH-11-11′), 7.18–7.14 (m, 2H, CH-10-10′), 7.11 (dd, *J* = 15.6, 8.1 Hz, 1H, CH-16), 5.72 (s, 1H, CH-5), 4.20 (q, *J* = 7.1 Hz, 2H, CH_2_-7), 1.20 (t, *J* = 7.1 Hz, 3H, CH_3_-8). ^13^C NMR (75 MHz, CDCl_3_) δ 165.04 (C2), 162.82 (C6), 156.73 (C4), 136.13 (C13), 134.50 (C12), 133.91 (C9), 129.24 (C15-15′), 129.04 (C11-11′), 129.01 (C10-10′), 126.22 (C16), 122.44 (C14-14′), 112.92 (C3), 61.54 (C5), 61.00 (C7), 14.12 (C8). MS, *m*/*z* (DART +): 358.08585, Anal. Calcd for ^12^C_19_^1^H_17_^35^Cl_1_^14^N_1_^16^O_4_.

*Ethyl 1-(4-bromophenyl)-4-hydroxy-5-oxo-2-phenyl-2,5-dihydro-1H-pyrrole-3-carboxylate* (DEXT-3). Yield (85%) as a pale-yellow solid, m.p. = 144–147 °C. HPLC purity: 99%. ^1^H NMR (300 MHz, CDCl_3_) δ 7.39 (d, *J* = 8.7 Hz, 2H, CH-14-14′), 7.36 (d, *J* = 8.5 Hz, 2H, CH-15-15′), 7.25 (dd, *J* = 12.7, 7.6 Hz, 3H, CH-11-11′, CH-12), 7.20 (d, *J* = 7.1 Hz, 2H, CH-10-10′), 5.68 (s, 1H, CH-5), 4.18 (q, 7.1 Hz, 2H, CH_2_-7), 1.17 (t, *J*= 6.9 Hz, 3H, CH_3_-8). ^13^C NMR (75 MHz, CDCl_3_) δ 165.06 (C2), 162.87 (C6), 156.37 (C4), 135.46 (C13), 134.78 (C9), 132.03 (C15-15′), 128.74 (C11-11′), 128.73 (C12), 127.46 (C10-10′), 123.43 (C14-14′), 118.94 (C16), 113.28 (C3), 61.40 (C5), 61.35 (C7), 13.91 (C8).

*Ethyl 1-(4-fluorophenyl)-4-hydroxy-5-oxo-2-phenyl-2,5-dihydro-1H-pyrrole-3-carboxylate* (DEXT-4). Yield (70%) as a white pale solid, m.p. = 157–160 °C. HPLC purity: 96%. ^1^H NMR (300 MHz, CDCl_3_) δ 7.46–7.31 (m, 2H, CH-14-14′), 7.29–7.19 (m, 2H, CH-11-11′), 7.14 (d, *J* = 8.4 Hz, 2H, CH-10-10′), 6.97 (t, *J* = 8.6 Hz, 2H, CH-15-15′), 5.65 (s, 1H, CH-5), 4.20 (q, *J* = 7.1 Hz, 2H, CH_2_-7), 1.19 (t, *J* = 7.1 Hz, 3H, CH_3_-8). ^13^C NMR (75 MHz, CDCl_3_) δ 165.03 (C2), 162.18 (C6), 160.54 (d, *J* = 246.9 Hz, C16), 156.69 (C3), 156.84 (C4), 134.65 (C12), 133.69 (C9), 132.15 (d, *J* = 2.9 Hz, C13), 129.05 (C11-11′), 128.88 (C10-10′), 124.45 (d, *J* = 8.2 Hz, C14-14′), 116.15 (d, *J* = 22.7 Hz, C15-15′), 61.56 (C5), 61.36 (C7), 14.12 (C8). MS, *m*/*z* (DART+): 376.07514, Anal. Calcd for ^12^C_19_^1^H_16_^35^Cl_1_^19^F_1_^14^N_1_^16^O_4_.

HPLC was performed in Agilent Liquid Chromatograph equipped with a Waters 2996 diode array. A column Synergi Polar-RP 80A 150 × 2.0 mm 4 um was used. Methanol was the initial eluent and water after 30 min. Flow = 0.2 cm^3^/min, wavelength 254 or 240 nm, sample solvent: acetonitrile or methanol.

### 5.2. Crystal Data for DEXT-2

C_19_H_17_ClNO_4_ M = 357.78, a = 9.8563(2) Å, b = 11.1958(3) Å, c = 16.5737(4)Å, α = 80.5940(10)◦, β = 74.5680(10)◦, γ = 84.1540(10)◦, V = 1735.98(3) Å^3^, T = 298(2) K, space group P-1, Z = 4, µ(Cu Kα) = 2.154 mm^−1^. A total of 22,795 reflections were measured, of which 6264 were independent (Rint = 0.0319). The final anisotropic full-matrix least-squares refinement on F^2^ with 717 variables converged at R1 = 6.09%, for the observed data and wR2 = 17.71% for all data. The goodness of fit was 1.158. Deposition Number CCDC 2121019 (https://www.ccdc.cam.ac.uk, accessed on 1 November 2021).

### 5.3. Bacterial Growth in the Presence of DEXT 1–4

The *P. aeruginosa* PA14 strain was kindly provided by Dr. Ausubel from the Department of Genetics, Harvard Medical School, Boston and was used as reference strain [29]. The INP-42 and INP-57M strains were provided by Dr. Rafael Coria Jimenez from the National Pediatric Institute, and both are clinical isolates from cystic fibrosis patients, but only INP-42 is multidrug-resistant [11]. The bacteria were kept in glycerol and frozen at −70 °C. 

The cultures were grown overnight (37 °C, 200 r.p.m.), adjusted to an O.D._600nm_ = 1, and the pyrrolidones were added at final concentrations of 10, 50 and 100 µM. They were incubated for 5 h at 37 °C at 200 r.p.m. and growth was measured every two hours.

### 5.4. Inhibition of Virulence Factors

The alkaline protease assay was performed as previously reported [30], with slight modifications. Briefly, the cultures (5 mL) were centrifuged at 13,000 r.p.m. for 1 min, and the supernatants were taken and filtered with 0.45 µm filters. Then, 5 mg of Hide Remazol Brilliant Blue R (Sigma) were added in 1.5 mL micro-centrifuge tubes and 875 µL of reaction buffer (20 Mm Tris-HCl pH 8.0, 1 mM CaCl_2_) and 125 µL of supernatant were added. The tubes were incubated for 1 h at 37 °C with 200 r.p.m. shaking. The samples were centrifuged at 13,000 r.p.m. for 5 min and the supernatant was taken. The absorbance was determined at 595 nm. The data were normalized with respect to bacterial growth. For all the assays, at least three independent cultures were used.

Pyocyanin was determined as described previously by Essar et al. (1990) [31]. Bacterial cultures were centrifuged for 3 min and 800 µL of supernatants were collected and vortexed with 420 µL of chloroform for 2 min. The samples were centrifuged for 5 min at 13,000 r.p.m. and the organic phase was collected (blue layer) and added to 800 µL of 0.1 N HCl. The mixture was vortexed for 1 min. Then, 650 µL of the aqueous phase (pink layer) were taken and 650 µL of distilled water were added. The pyocyanin was determined at 520 nm. The data were normalized with respect to bacterial growth. For all the assays, at least three independent cultures were used.

#### 5.4.1. *P. aeruginosa* Biofilm Formation

Overnight cultures of the selected bacteria were diluted in fresh LB medium to an O.D._600nm_ = 0.05 and 200 µL per well were deposited onto a plate of 96 wells (corning^®^) and the pyrrolidones were added at final concentrations of 10, 50 and 100 µM. The plate was incubated for 24 h without shaking at 37 °C. Later, the biofilm formation was quantified as reported previously [32] with slight modifications. Briefly, the cultures were discarded, and the plate was washed two times with water to remove the remaining planktonic cells. Subsequently, 200 µL of violet crystal dye at 0.1% was added and left for 15 min. The excess of dye was removed by rinsing with distilled water. Later, 200 µL of acetic acid 30% was added and left for 15 min. The crystal violet concentration was measured in a spectrophotometer O.D._570nm_ using acetic acid 30% as blank. The data were normalized with respect to bacterial growth. Each assay was performed three independent times with nine replicates.

#### 5.4.2. Swarming Motility

M9 minimal medium supplemented with 1 mM MgSO_4_, 0.5% glucose, 0.5% casamino acids, and 0.5% agar was used for motility assays [33]. Three mL per well were added in 6-well plates (Corning^®^) with the pyrrolidones at 100 or 50 µM. Aliquots (2.5 µL) from overnight cultures were loaded at the center of each well. The migrations zones were measured after 24 h of incubation at 37 °C. For each assay, three independent cultures were used.

### 5.5. Type III Secretion Assay

The type III secretion assay was performed as previously reported [34]. Briefly, *P. aeruginosa* strains from overnight cultures were grown (1:200 dilution) in 4 mL of LB supplemented with 10 mM MgCl_2_, 0.5 mM CaCl_2_ and 5 mM EGTA (pH 7.4) in the presence of 100 µM DEXT 1–4 or 50 µM MBX-1641, at 37 °C with aeration until they reached an O.D._600nm_ of 0.8–1.0. 1 mL of each culture was centrifuged at 18,000× *g* for 2 min and the bacterial cell pellet was frozen. The supernatant was transferred to another tube and centrifuged again to remove the remaining bacteria. Proteins were precipitated from the supernatant with 10% *v*/*v* trichloroacetic acid (TCA) at 4 °C for 12 h and centrifuged at 18,000× *g* for 30 min. The bacteria and protein pellets were resuspended in SDS-PAGE loading buffer supplemented with 1% *v*/*v* β-mercaptoethanol. The volume was normalized according to the O.D. of each culture. Residual TCA from protein pellets was neutralized by adding 10% *v*/*v* saturated Tris. Samples were boiled for 5 min and loaded onto 15% SDS-polyacrylamide gels under denaturing conditions. Proteins were transferred to a nitrocellulose membrane. The membrane was blocked and probed against anti-ExoU and anti-ExoS polyclonal antibodies. The proteins were detected using the Immobilon Western Chemiluminescent HRP Substrate Kit (Merck, Darmstadt, Germany) on X-ray films.

### 5.6. Mouse Necrosis Model

CD-1 mice (equal proportion of males and females) were obtained from the Facultad de Estudios Superiores, Cuautitlán-UNAM. They were kept under standard conditions (23 °C ± 2 °C) with a 12 h light-dark cycle and free access to food (laboratory rodent diet 5001, LabDiet^®^) and drinking water. The mouse necrosis model was performed as previously reported [35,36]. Briefly, six weeks old CD1 male mice were depilated using a cream hair remover (Loquay^®^). The mice were anesthetized with an intraperitoneal injection of 64 mg/kg sodium pentobarbital (Pisabental, PiSa^®^). Before the injection, the cultures were grown to an O.D. _600nm_ ~1.0 in LB broth, and the bacterial cells were washed twice with sterile PBS and adjusted to 1 × 10^7^ CFU. After that, 60 µL of the bacterial suspension were injected subcutaneously into the right side of the dorsum. C-30, MBX-1641, and DEXT-3 at different concentrations were diluted in the bacterial suspension and injected into the subcutaneous space with the bacteria. PBS was used as a negative control. The necrotic lesion was measured every 24 h for four days.

After four days post-inoculation, the livers and the soft tissues containing the necrotic area of the mice were excised and homogenized with PBS. Serial dilutions were done in LB plates to count colony-forming units. Experiments were performed at least twice with five animals per group.

### 5.7. Ethical Declaration

Experiments with mice were carried out following the indications of the Research, Ethics, and Biosafety Committees of the Hospital Infantil de México-Federico Gómez (HIM2018-002. Date of approval 21 May 2018).

## Figures and Tables

**Figure 1 molecules-26-07635-f001:**
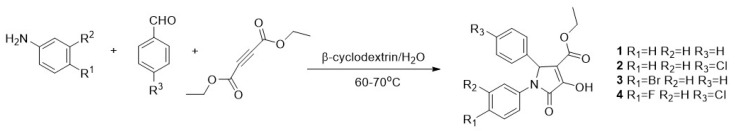
Synthesis of **1**–**4** halogen pyrrolidones. Conditions. To an aqueous solution of β-cyclodextrin an aniline derivative was added. Subsequently, diethyl acetylenedicarboxylate was added slowly followed by an aromatic aldehyde. The reaction mixture was heated at 60–70 °C.

**Figure 2 molecules-26-07635-f002:**
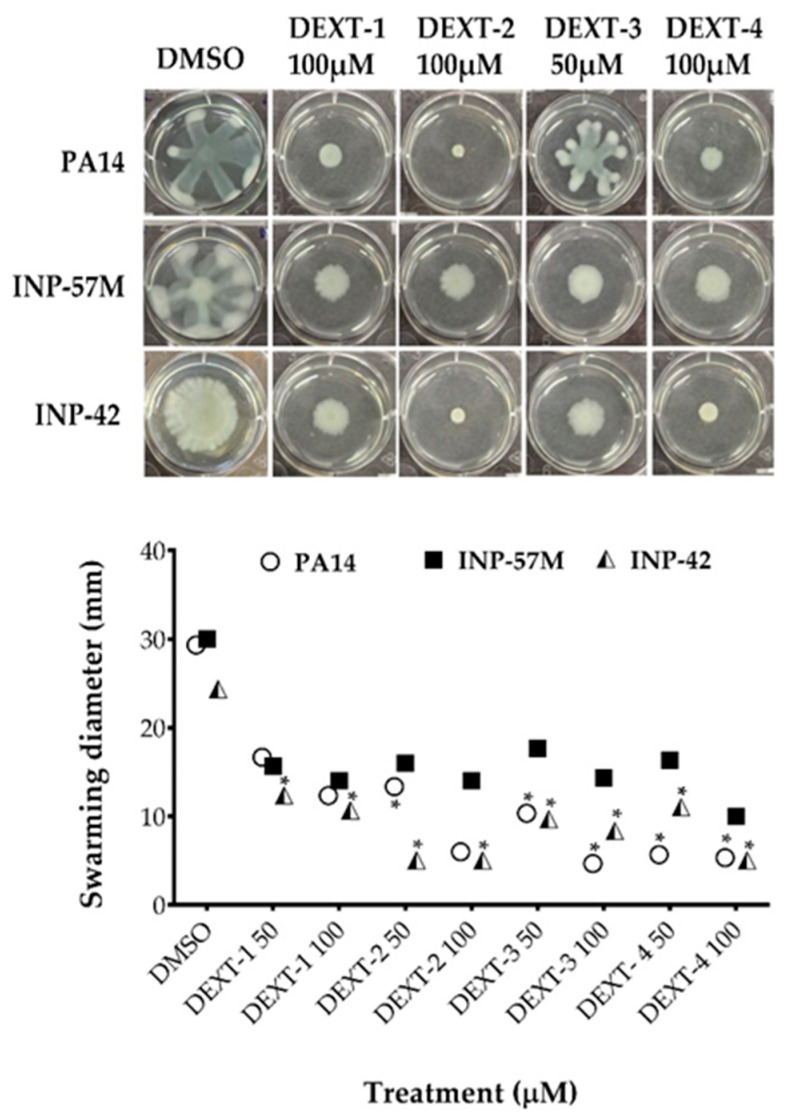
Effect of halogenated pyrrolidones on swarming motility of *P. aeruginosa* PA14 WT and clinical isolates (INP-57M and INP-42). Values are presented as means ± SD. The Student’s *t*-test was used to calculate the differences between two mean values (* *p* < 0.05).

**Figure 3 molecules-26-07635-f003:**
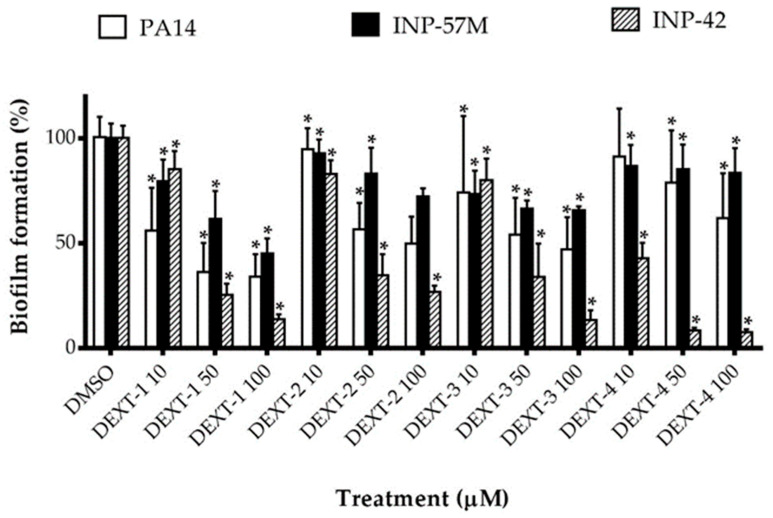
Effect of halogenated pyrrolidones on biofilm formation of *P. aeruginosa* PA14 WT and clinical isolates (INP-57M and INP-42). Values are presented as means ± SD. The Student’s *t*-test was used to calculate the differences between two mean values (* *p* < 0.05).

**Figure 4 molecules-26-07635-f004:**
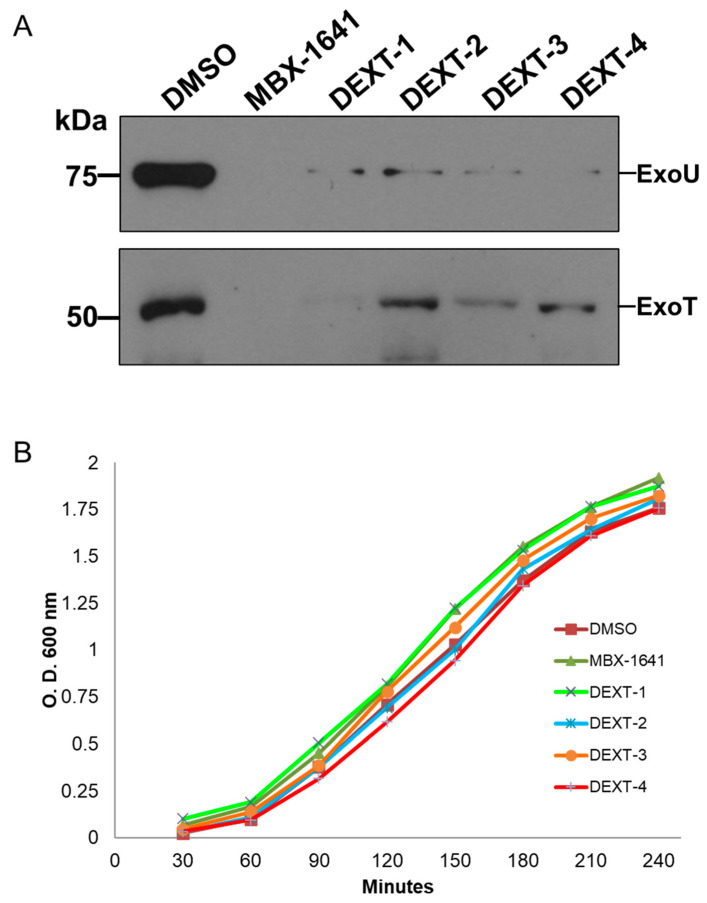
(**A**) Inhibition of the secretion of the effector proteins ExoU and ExoT by MBX-1641 (a T3SS inhibitor used as positive control) and the pyrrolidones DEXT 1–4 [100 μM]. Secreted proteins were immunodetected using specific antibodies against ExoU and ExoS (the ExoS polyclonal antibodies cross-react with ExoT). Control: DMSO (**B**) Growth curve showing no effect of MBX-1641 or DEXT 1–4 on the viability of *P. aeruginosa* PA14 WT. Control: DMSO.

**Figure 5 molecules-26-07635-f005:**
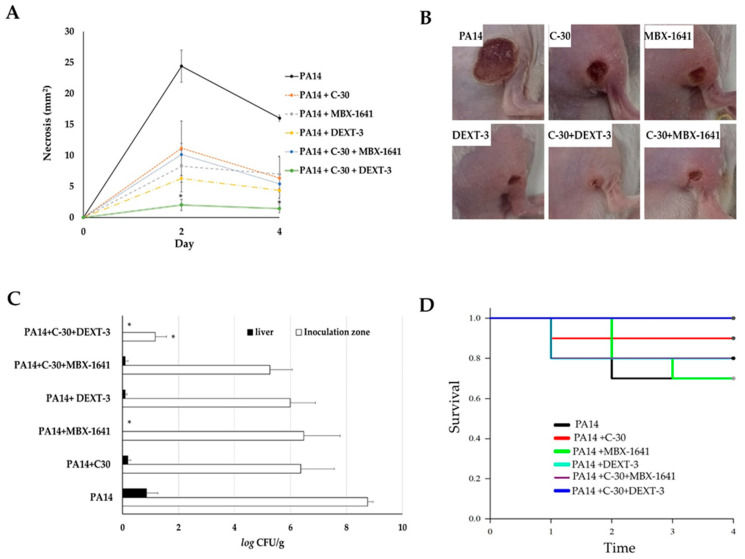
The antipathogenic activity of C-30 (QS inhibitor), MBX-1641 (T3SS inhibitor), DEXT-3, and C-30/DEXT-3 and C-30/MBX-1641 combinations in a model of infection and necrosis induced by *P. aeruginosa*. Mice intraperitoneally inoculated with *P. aeruginosa* PA14 WT strain (1 × 10^7^ CFU) were treated separately with C-30 (50 µM), MBX-1641 (200 µM), DEXT-3 (100 µM), and C-30 (50 µM) + MBX-1641 (200 µM), and C-30 (50 µM) + DEXT-3 (100 µM) combinations (**A**) Induction of necrosis. (**B**) Representative images of in situ damage. (**C**) Bacterial load in the inoculation area (CFU in tissue) and systemic dispersion (CFU in liver), * α = 0.05. (**D**) Kaplan-Meier survival curve, performed with the GraphPad Prisma 6 program (95% confidence interval) and the data analysis in the SPSS 22.0 program. The data are representative of two experiments (*n* = 5).

**Figure 6 molecules-26-07635-f006:**
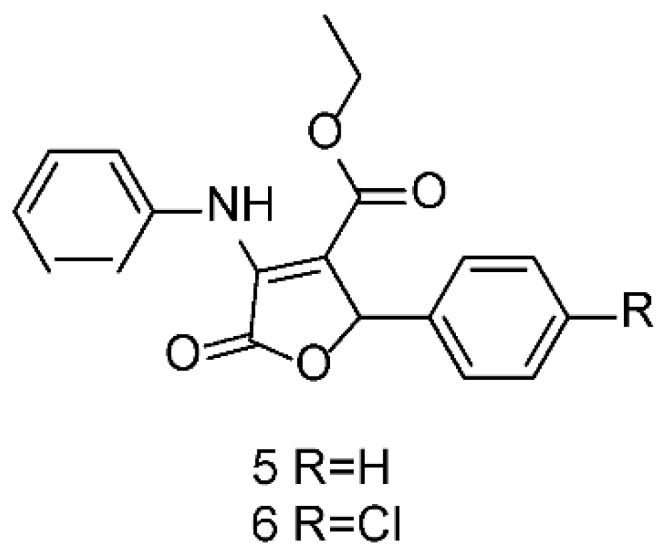
Structures of **5** and **6** furanones.

**Figure 7 molecules-26-07635-f007:**
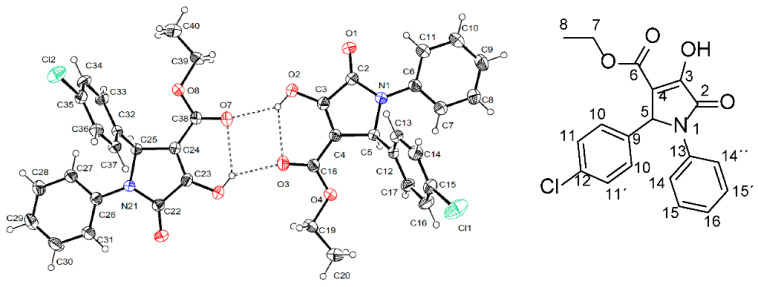
Stereo structure of DEXT-2 determined by X-ray crystallographic analysis.

**Table 1 molecules-26-07635-t001:** Reaction Time used in the synthesis of DEXT 1–4 pyrrolidones yield and melting point.

Entry	Reaction Time	Yield (%)	M.p. °C
1	12 h	80	152–153
2	16 h	90	165–167
3	18 h	85	144–147
4	12 h	70	157–160

## Data Availability

Not applicable.

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
