# Peer review of "Anti-Pathogenic Properties of the Combination of a T3SS Inhibitory Halogenated Pyrrolidone with C-30 Furanone"

_molecules, 2021, doi:10.3390/molecules26247635_

Round 1
Reviewer 1 Report
1)Introduction: This chapter could be improved to enhance the objective of this research work. Please do note ‘furthermore’ was used too many times throughout Introduction. Suggestion of alternative words choice, eg. besides, additionally, in addition, etc..
2) The diagrams in Figure 4 need huge improvement. The details in graphs for (A), (C) and (D) could hardly be seen at all. (B) size need to be enlarged. This is of importance as Section 2.4 is dependent on Figure 4.
3)The molecular structure in Figure 6 is not clear.
4)Please recheck line 189 on the quoting of reference 13.
5)Line 205,Pg 7: Is P. aeruginosa and Pseudomonas aeruginosa not the same?
6)Line 270, Pg 8: What is ‘eluyent’?
7)Section 4.1.1: The NMR assignments for the synthesized compounds were given in details. However, I have no idea on the assignment(s) as there were no numberings given in the molecular structure in journal article and supplementary materials.
8)Section 4.1.1: Line 311, Pg 9: typo in unit
9)Section 4.2, Line 322, Pg9: error in formatting
10)Section 4.3, Line 331, Pg9: error in formatting
11)I am not sure how the authors would like to arrange for Line 332 and 333 (Pg9).
12) Conclusion needs a 'Huge improvement' to showcase the significance and relevance of the research work.
13) Reference section is messy. I believe that there should be 28 references instead of the listed 31 references. Authors need to recheck carefully. However, 28 references might not be sufficient for this research paper. I would recommend the authors to include more references to support the Introduction and Discussion write-up.
14) Reference 13 is not quoted correctly and needs to be check carefully.
15) Supplementary Materials: Labelling in the NMR spectra needs to be rework carefully.
Author Response
Response to Reviewer 1 Comments
Point 1: Introduction: This chapter could be improved to enhance the objective of this research work. Please do note ‘furthermore’ was used too many times throughout Introduction. Suggestion of alternative words choice, eg. besides, additionally, in addition, etc.
Response 1: The correction was done, and the added information is highlighted in yellow in the document.
Point 2: The diagrams in Figure 4 need huge improvement. The details in graphs for (A), (C) and (D) could hardly be seen at all. (B) size need to be enlarged. This is of importance as Section 2.4 is dependent on Figure 4. Point 3: The molecular structure in Figure 6 is not clear.
Response 2 and 3: Suggested modifications were made. Also, the images with the quality and format changes were attached in separate files on the magazine platform.
Point 4: Please recheck line 189 on the quoting of reference 13.
Response 4: More references were appended, and errors in order and format were carefully checked and corrected.
Point 5: Line 205, Pg 7: Is P. aeruginosa and Pseudomonas aeruginosa not the same?
Response 5: It is the same bacteria. It was corrected, abbreviating the genus as P. aeruginosa.
Point 6: Line 270, Pg 8: What is ‘eluyent’?
Response 6: the correct word is “eluent”.
Point 7: Section 4.1.1: The NMR assignments for the synthesized compounds were given in details. However, I have no idea on the assignment(s) as there were no numberings given in the molecular structure in journal article and supplementary materials.
Response 7: The numbered structure was added
Point 8: Section 4.1.1: Line 311. Pg 9: typo in unit. Point 9: Section 4.2, Line 322, Pg9: error in formatting. Point 10: Section 4.3, Line 331, Pg9: error in formatting.
Response 8-10: The correction was done, and the added information is highlighted in yellow in the document.
Point 11: I am not sure how the authors would like to arrange for Line 332 and 333 (Pg9).
Response 11: The materials and methods section were restructured. Changes are highlighted in yellow.
Point 12: Conclusion needs a 'Huge improvement' to showcase the significance and relevance of the research work.
Response 12: The conclusions were rewritten to make them more precise and more concise.
Point 13: Reference section is messy. I believe that there should be 28 references instead of the listed 31 references. Authors need to recheck carefully. However, 28 references might not be sufficient for this research paper. I would recommend the authors to include more references to support the Introduction and Discussion write-up. Point 14: Reference 13 is not quoted correctly and needs to be check carefully.
Response 13-14: More references were appended, and errors in order and format were carefully checked and corrected.
Point 15: Supplementary Materials: Labelling in the NMR spectra needs to be rework carefully.
Response 15: Since trying to assign most of the signals in the small spaces of the spectra was confusing, it was decided to simplify the assignments and make the reading easier.

Reviewer 2 Report
The manuscript is well written, well organized and presents promising results on discovering new drugs with anti-virulence action on the widespread gram-negative bacterium Pseudomonas aeruginosa. The reviewer suggested some changes in the manuscript in order to be publicly acceptable in Molecules.
1. Introduction
- Please, write in full what CFV means.
- In the last sentence, add a comma after “In this work”.
2. Results
- The legend of figure 1 must be more detailed in order to give a better idea about the synthesis process as whole.
- The authors said “The pyrrolidones (DEXT 1-4) did not show a bactericidal effect at 10-, 50-, and 100-µM concentrations on the INP-42 strain; however, DEXT 3 at 100 µM induced a slight growth delay in PA14 reference and INP-57M strains (data not shown). Similarly, DEXT 1-4 compounds did not significantly reduce pyocyanin production. Though, DEXT-1, DEXT-2, and DEXT-4 inhibited protease activity by around 10 to 30% in the two clinical isolates INP-42 and INP-57M (data not shown).” The reviewer suggests separating the anti-proliferative to the anti-virulence results, because the sub item 2.2 is entitled “Anti-virulence properties…” In this context, the reviewer suggest to create a previous item showing the absence of action on the P. aeruginosa growth (please, it is very relevant and interesting to demonstrate the results instead of describe them as “data not shown”).
- About the figure 2, the authors firstly described the results refereeing to the figure 2B and then the results referring to the figure 2A. Please, change the order of the figure 2 descriptions. Alternatively, the reviewer strongly suggest, in order to facilitate the visualization of all these initial experiments, that the author split the figure 2 in two distinct figures: one of them (the first one) describing the swarming (and also the negative results on the production of both pyocyanin and protease) and the other to show the biofilm results. Please, add more methodological details in the legend of figure 2, for instance, describe the statistical test which was applied on these results.
- Item 2.3: The authors said “The type III protein secretion assay was performed as previously reported.” This sentence is completely vague. The authors should remember that methodology is placed after the results’ section. So, the authors should explain, in this case, how the experiment was performed even if in a concise way. Also, legend of figure 3 should be more detailed to help the reader to understand the results as a whole, even without the necessity to consult the methodology section.
3. Methodology
- In the sub item “Strain”, it is relevant that the authors associate the strain code to a bacterial species, so change the sentence to “The PA14 strain ...” to “The P. aeruginosa PA14 strain…” Moreover, are the three P. aeruginosa strains used in the present work classically MDR strains? Please, clarify this issue!
- In the sub item “Bacteria growth in the presence of DEXT 1-4”, why the authors did not use the standard protocol of CLSI to measure antimicrobial action of the test drugs?
Author Response
The manuscript is well written, well organized and presents promising results on discovering new drugs with anti-virulence action on the widespread gram-negative bacterium Pseudomonas aeruginosa. The reviewer suggested some changes in the manuscript in order to be publicly acceptable in Molecules.
Point 1: Please, write in full what CFV means. Point 2: In the last sentence, add a comma after “In this work”.
Response 1 and 2: The correction was done (line 316), and the added information is highlighted in yellow in the document.
Point 3: The legend of figure 1 must be more detailed in order to give a better idea about the synthesis process as whole.
Response 3: Some experimental data were added in the legend of figure 1,
Point 4: The authors said “The pyrrolidones (DEXT 1-4) did not show a bactericidal effect at 10-, 50-, and 100-µM concentrations on the INP-42 strain; however, DEXT 3 at 100 µM induced a slight growth delay in PA14 reference and INP-57M strains (data not shown). Similarly, DEXT 1-4 compounds did not significantly reduce pyocyanin production. Though, DEXT-1, DEXT-2, and DEXT-4 inhibited protease activity by around 10 to 30% in the two clinical isolates INP-42 and INP-57M (data not shown).” The reviewer suggests separating the anti-proliferative to the anti-virulence results, because the sub item 2.2 is entitled “Anti-virulence properties…” In this context, the reviewer suggest to create a previous item showing the absence of action on the P. aeruginosa growth (please, it is very relevant and interesting to demonstrate the results instead of describe them as “data not shown”).
Response 4: All the suggested changes were made and the information referring to the growth curves (Figure S1) and anti-virulence evaluation (Figure S2) was annexed in the supplementary material.
Point 5: About the figure 2, the authors firstly described the results refereeing to the figure 2B and then the results referring to the figure 2A. Please, change the order of the figure 2 descriptions. Alternatively, the reviewer strongly suggest, in order to facilitate the visualization of all these initial experiments, that the author split the figure 2 in two distinct figures: one of them (the first one) describing the swarming (and also the negative results on the production of both pyocyanin and protease) and the other to show the biofilm results. Please, add more methodological details in the legend of figure 2, for instance, describe the statistical test which was applied on these results.
Response 5: The figure's order was changed and divided as suggested by the reviewer. Also, the requested information (growth curves and virulence factors) was added to supplementary material, and the statistic information was used.
Point 6: Item 2.3: The authors said “The type III protein secretion assay was performed as previously reported.” This sentence is completely vague. The authors should remember that methodology is placed after the results’ section. So, the authors should explain, in this case, how the experiment was performed even if in a concise way. Also, legend of figure 3 should be more detailed to help the reader to understand the results as a whole, even without the necessity to consult the methodology section.
Response 6: The correction was done, and the added information is highlighted in yellow in the document (lines 285-298).
Point 7: Methodology: In the sub item “Strain”, it is relevant that the authors associate the strain code to a bacterial species, so change the sentence to “The PA14 strain ...” to “The P. aeruginosa PA14 strain…” Moreover, are the three P. aeruginosa strains used in the present work classically MDR strains? Please, clarify this issue!
Response 7: Suggested changes were made. The information was attached on lines 370 to 371, and all changes are highlighted in yellow in the text. Of the three strains used in this research, only the INP-42 strain is drug-resistant.
Point 8: In the sub item “Bacteria growth in the presence of DEXT 1-4”, why the authors did not use the standard protocol of CLSI to measure antimicrobial action of the test drugs?
Response 8: The CLSI protocols are designed primarily to make comparisons of bactericidal effectiveness between antibiotics. The main interest in our trials is the effect at subinhibitory concentrations (doi.org/10.3389/fmicb.2021.667126), where the anti-virulence phenomenon occurs. The concentrations and conditions used are based on previous studies of the anti-virulence activity of furanones (DOI: 10.1093/femspd/ftv040; DOI: 10.1099/00221287-148-1-87 and DOI. https://doi.org/10.3390/molecules26061620).

Reviewer 3 Report
The study sounds good and important to readers
-It is preferable to write the method of detecting swarming motility and pyocyanin and rhamnolipids production and type III protein secretion assay.
-The study includes animals. Housing criteria, ethical approval no. and date should be included?
Author Response
Point 1: It is preferable to write the method of detecting swarming motility and pyocyanin and rhamnolipids production and type III protein secretion assay.
Response 1: The correction was done, and the added information is highlighted in yellow in the document. (Lines 377-418)
Point 2: The study includes animals. Housing criteria, ethical approval no. and date should be included?
Response 2: The requested information is shown in lines 440 to 443, also in the paragraph of lines 459 to 461.
